# A Cluster Randomized Controlled Trial of the Dental Sealants Quality in Rural Schoolchildren Using Innovative Suction without Dental Assistance

**DOI:** 10.3390/ijerph20054035

**Published:** 2023-02-24

**Authors:** Songchai Thitasomakul, Sukanya Tianviwat

**Affiliations:** Evidence-Based Dentistry for Oral Health Care and Promotion Phase II Research Unit, Preventive Department, Faculty of Dentistry, Prince of Songkla University, Songkhla 90110, Thailand

**Keywords:** dental sealant, retention, caries, dental nurses, innovation, moisture control innovation

## Abstract

This study examined whether the moisture control innovation (tongue and cheek retractors and saliva contamination (SS-suction)) used without dental assistance could improve the quality of dental sealant in rural Thai school children compared to a standard treatment, i.e., high power suction with dental assistance. A single blind, cluster randomized controlled trial was conducted. Participants were 15 dental nurses working in sub-district health promoting hospitals and 482 children. All dental nurses attended workshops of SS-suction and revised dental sealant procedures. Children with sound lower first permanent molar teeth were simple-randomly assigned to either an intervention or control group. The children in the intervention group were sealed with SS-suction, and the children in the control group were sealed with high power suction and dental assistance. There were 244 children in the intervention group and 238 children in the control group. Dental nurses’ satisfaction on SS-suction was record by visual analogue scale (VAS) for each tooth during treatment. After 15–18 months, caries on sealed surfaces were examined. The results showed that the median satisfaction score of SS-suction was 9 out of 10, and 17–18% children experienced uncomfortable sensation during insertion or removal. The uncomfortable feeling disappeared once the suction was in place. Caries on sealed surfaces did not differ significantly between the intervention and control groups. Caries on the occlusal surface was present in 26.7% and 27.5%, and caries on the buccal surface was present in 35.2% and 36.4% of cases in the intervention and control groups, respectively. In conclusion, dental nurses were satisfied with SS-suction in terms of both function and safety. The effectiveness of SS-suction was compatible with the standard procedure after 15–18 months.

## 1. Introduction

According to the World Health Organization (WHO), the ultimate goal of service delivery is to provide high-quality care in a fair manner [1]. Oral health inequity remains a major problem in several countries, including Thailand, not only in terms of accessibility, but also in terms of oral health [2,3]. To improve accessibility, dental services have been established in sub-district health promoting hospitals (SHPH) in Thailand and dental nurses have been assigned to provide preventive care to children. However, due to limited resources, the presence of dental assistants in the Thai oral health system is insufficient [4]. There are 3584 dental nurses, while only 213 dental assistants work at the sub-district level [5]. Dental caries continues to be a major problem. The most recent national oral health survey in Thailand [6] found that 52.0% of 12-year-old children suffer from dental caries. Therefore, the dental sealant program in schools recommended by the American Association of Community Dental Programs [7] has been implemented nationally in Thailand since 2005. However, this program struggled with quality issues [8,9,10], especially in the Thai context, of short-term retention related to moisture control problems [8,9]. Recently, an innovation (tongue and cheek retractor and saliva contamination (SS-suction)) was invented to improve the quality of sealing by increasing the quality of moisture control, especially for the lower jaw under the shortage of dental assistants. The SS-suction is a universal design and consists of three parts (Figure 1). The aspirator tips are covered with silicone tubes that reach the buccal and lingual sides of a tooth. The chin plate is pressed against the chin and fixes the suction in place. The last part is the connecting tube, which is connected to the saliva tube of the dental chair (Figure 1, Figure 2 and Figure 3) [11]. This innovation was invented by the authors (ST and ST) and tested for safety both in the laboratory and in a pilot trial on school children [11,12]. The aim of this study was to investigate the satisfaction of dental nurses and to compare the effectiveness of dental sealant (caries) under two conditions: SS-suction without dental assistance and high-power suction with dental assistance.

## 2. Materials and Methods

This study was conducted in a southern Thai province and included 15 eligible sub-district health promoting hospitals (SHPHs) in rural areas that performed dental sealants in schools. The study design was a single blind (outcome assessors) cluster randomized control trial and parallel group study with balanced randomization (1:1). The clusters were SHPHs that included groups of dental nurses who implemented school dental sealant programs. The cluster study was designed because the children were treated by responsible dental nurses in each SHPH. The children included according to the criteria: at least one caries-free lower first permanent molar with deep pits and fissures, and good cooperation. The sealant material used in this study was Concise^®^ (3M). The children in the intervention group were sealed with SS-suction, and the children in the control group were sealed with high power suction and dental assistance (DA). At the beginning of the study, a workshop was conducted to explain the objectives of the program, the use of SS-suction, the collection of satisfaction data related to its use and safety, recording child data, and the revised dental sealing technique. After the workshop, each dental nurse received ten SS-suctions and then the school sealant program was conducted in the usual manner. Children’s baseline data included gender, age, toothbrushing behavior, use of fluoride toothpaste, and sugar consumption. Dental characteristics included cooperation during service, position, and oral hygiene. Satisfaction with use included tongue retraction, buccal retraction, insertion and removal, attachment, workspace obstruction, sucking, child cooperation, and overall satisfaction. Satisfaction measured by VAS when 0 was unsatisfactory and 10 was very high satisfaction. Safety aspects included pain and ulceration of the chin and mouth. All data were presented at both the dental nurse and child levels.

The outcome as an indicator of school sealant quality was the presence of caries on the sealed surfaces. The outcome was assessed at 15–18 months after sealing. After the start of the study, the results and methods were changed from 6 and 12 months to 15–18 months due to the COVID-19 pandemic situation. The evidence of caries or the final outcome was discoloration of the occlusal surface and sticking with gentle probing according to Simonsen criteria [8,13]. There were two examiners (authors) blind to the children’s group and calibration of the examination was performed. The kappa values for intra-examiner agreement were 0.82 and 0.91, and the inter-examiner agreement was 1.00. Baseline variables were collected by the dental nurses to determine the characteristics of the teeth and the children and were noted in the treatment forms. Sample size was calculated based on the guidelines for cluster randomized trials proposed by Cosby et al. [14] and was calculated using the following values from a previous study [15]: Intraclass correlation = 0.012, power = 80%, outcome difference between intervention and control groups = 8%, and alpha = 0.05. The number of clusters was 15, and the total estimated number of children needed was 220 for the intervention group and another 220 for the control group. Each dental nurse provided services to both groups based on simple random assignment using a computer-generated list of random numbers (by the research manager). An allocation assignment envelope was opened and announced when the children were seated in the dental chair just before sealing.

Data analysis was performed using the program R, version 4.2.1 (R Development Core Team, Vienna, Austria) [16]. Multilevel modeling was used for hierarchical structures and clustered data [17]. Caries on the sealed surfaces at the follow-up time were the dependent variables because all sealed teeth began caries free, nested at lower level. In the data analysis, caries were divided into two models: occlusal and buccal surfaces, since SS-suction may have different effects on the different sides. Adjustment for cluster effects was performed by including children and dental assistants as higher levels in the analysis. In the final model, the variables of interest were group (control or intervention), children’s age, tooth position and follow-up time. There was no significant change in the method after the study began, and no interim analysis was performed. An intention-to-treat (ITT) was analyzed for all follow-up (FU) cases and sensitivity analysis was the best-case and worst-case scenarios [18]. The best-case scenario was all children with missing outcomes in SS-suction group had good outcome (no caries), and all those with missing outcome in DA group had poor outcome (caries). The worst-case scenario was the converse.

### Statement of Approval for Human Subjects and Registry for the Clinical Trial

The research protocol was approved by the Ethics Committee of the Faculty of Dentistry, Prince of Songkla University (EC6401-004). This study was registered in the Thai Clinical Trials Registry (TCTR) under study number TCTR20210401007.

## 3. Results

Fifteen dental nurses who enrolled in this study had a mean age of 32 ± 7 years and 10 ± 7 years of experience. At baseline, the number of children and teeth sealed in the control group was 238 children with 348 teeth and in the intervention group was 244 children with 324 teeth. At 15–18 months, there were 205 children with 313 teeth in the control group and 205 with 273 teeth in the intervention group (Figure 4). The drop-out rate was 14% of children in the control group and 16% in the intervention group. The main reason for this was refusal to undergo the examination due to COVID-19 awareness. The social and behavioral characteristics of the children and teeth between the two groups did not differ at baseline but significant difference at follow-up time were evident (Table 1 and Table 2).

Satisfaction with the use of SS-suction and safety is shown in Table 3. The median satisfaction with SS-suction was 9 and the interquartile range was between 8–10 in all categories. Safety aspects were divided into discomfort or pain and ulceration in two areas: chin and mouth. Dental nurses reported discomfort on left and right sides in the chin (6.7, 8.3%) and discomfort in the mouth (16.7, 18.1%) while inserting or removing suction. The discomfort was due to pressure in the chin and mouth. After insertion and stable suction, this feeling ceased. In this study, 1.1–3.3% of ulcers were reported by dental nurses presenting as red areas on the soft tissue and recovering after the removal of suction within 5–10 min of observation. Table 4 shows oral hygiene and caries on the occlusal and buccal surfaces of the sealed teeth, which did not differ significantly between groups. Nearly 60% of children in both groups had poor oral hygiene, and caries incidence was similar in both groups at 27–28% on the occlusal surfaces and 35–36% on buccal surfaces. The multilevel logistic regression models for occlusal caries and buccal caries of FU cases (Table 5 and Table 6) and best-case and worst-case scenarios showed no significant differences between the groups when controlling for tooth position, age, and follow-up time. The best-case and worst-case scenarios presented similar, non-significant results and the adjusted odd ratios (OR) of SS-suction to prevent caries on sealed surfaces were 1.15–1.22 times greater than the DA group.

## 4. Discussion

High satisfaction and safety regarding SS-suction were reported among dental nurses even though the service is provided without dental assistants. Caries as reported outcomes of children sealed with SS-suction were comparable to those of children sealed with the usual standard procedure, i.e., high-power suction by dental assistants.

Satisfaction with SS-suction was very high in all dimensions and helped dental nurses to provide dental services without dental assistance, which was consistent with the previous efficacy study [19]. However, the discomfort and red spots caused by the pressure of the innovation could be related to the jaw size. The SS-suction incorporates a universal design that is easy to use due to a size-adjustable chin plate with a coil spring. An efficacy study in another province showed a much lower percentage of uncomfortable areas on the chin (3.7%) and mouth (5.4%), while only one in 266 cases had an ulcer on the chin. However, the body size of the children was different. In this study, the dental nurses complained that the SS-suction did not fit the children’s jaws. The percentage of normal growth (weight and height) from the Thailand Heath Data Center (HDC) of 2021 [20] in the previous SS-suction study was 51.4% compared to the current study of 39.8. In this study, the caries rate on sealed surfaces was higher than in the international studies and other Thai studies [21,22]. A meta-analysis of the new caries among sealed teeth in Thailand in the period of 12–24 months on occlusal surfaces was 17.6 ± 4.9% [23] compared to 27% in this study. This is due to the high sealant loss [8] and high caries risk in this region [24].

To evaluate the quality of moisture control, short-term retention is a good indicator. However, the COVID-19 pandemic during the research led to lifestyle changes in Thailand. The government declared a state of emergency and ordered the closure of educational institutions throughout the country. The evaluation period was delayed to 15–18 months and showed low complete retention of 31–32% on the occlusal and 17–18% on buccal surfaces [25]. In a previous cohort study of dental sealants in Thailand [3], the sealed teeth were examined every six months for 2.5 years. The study showed that the majority of sealant retention lost dramatically in the first 6 months (32.9%), with a gradual decrease in rate of sealant loss later. The sealant retention rate was a good indicator of the intermediate outcome when we made a short-term evaluation. Caries development, on the other hand, was the final outcome that we expected from the prevention program. With partial and complete loss of sealant, the caries prevention was not different from that in unsealed teeth [15,26], even though some studies claimed the effect of resin tag [27]. The efficacy study of SS-suction reported 79.4% of complete retention after three months [19], which was much higher than the meta-analysis of six-month sealant retention in Thailand of 53.8% [23]. However, there were no studies reporting retention at the same time of follow-up to compare with this efficacy study.

In this study, the interesting factor investigated was moisture control, and the controlled factors were age, tooth position, follow up time, dietary and toothbrushing behavior, and fluoride exposure. The oral health prevention and promotion program has been implemented in schools in Thailand for more than 30 years, but it was temporarily interrupted during the COVID-19 pandemic [28], and all schools remained closed for almost a year. Therefore, caries incidence was very high in both groups.

The strength of the study lies in the study design, and the ITT and sensitivity analysis were performed to confirm the result. A limitation of the study is the follow-up time, and the interim results were not assessed. Suggestions for further studies are to improve the design of the SS vacuums, especially their size, to consider the appropriate combination of toothbrushes and dental sealants, and conduct an economic evaluation of the innovation. Due to the property of aerosol reduction [12], SS-suction is the instrument of choice for use in dental clinics, regardless of whether or not dental assistance is inadequate.

## 5. Conclusions

In summary, dental sealant with SS-suction without dental assistance has equivalent effectiveness to high-power suction with dental assistance. However, the overall effectiveness of dental sealant needs to be improved.

## Figures and Tables

**Figure 1 ijerph-20-04035-f001:**
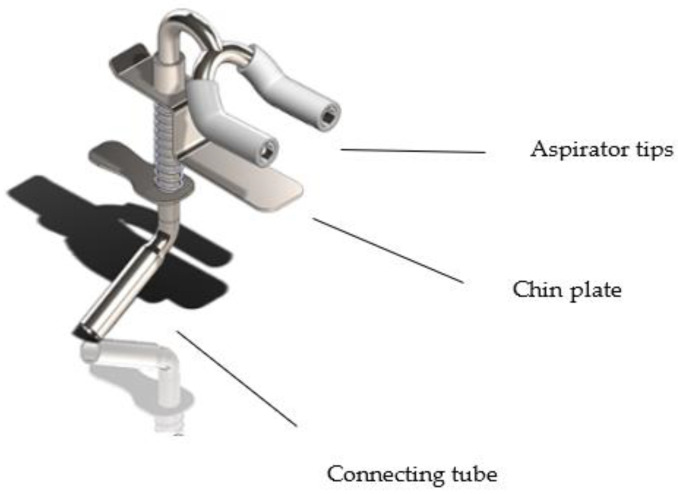
Components of SS-suction.

**Figure 2 ijerph-20-04035-f002:**
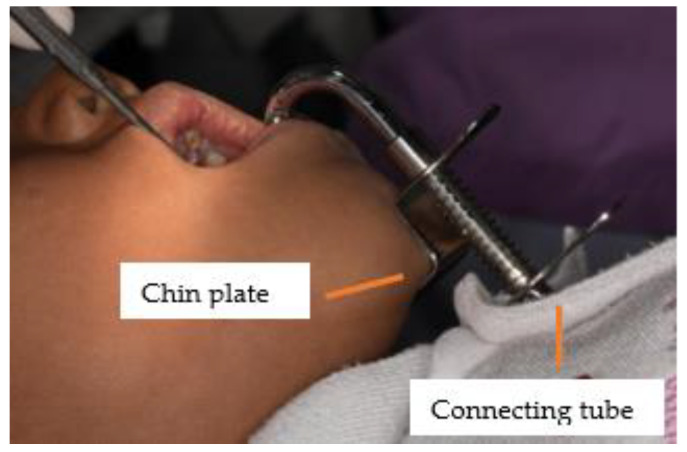
Extra-oral view of SS-suction in patients.

**Figure 3 ijerph-20-04035-f003:**
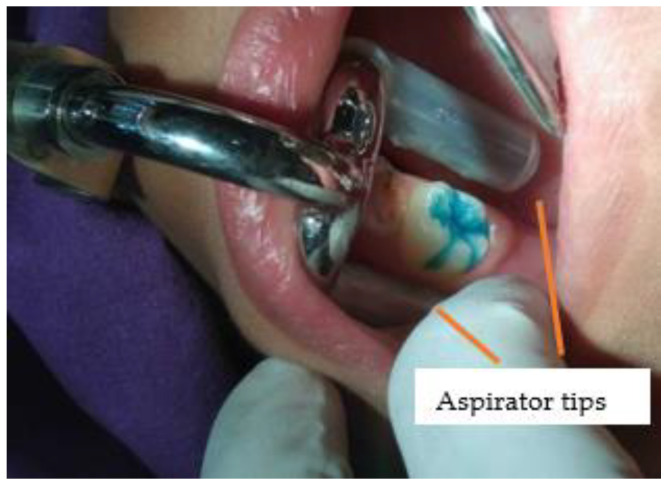
Intra-oral view of SS-suction in patients.

**Figure 4 ijerph-20-04035-f004:**
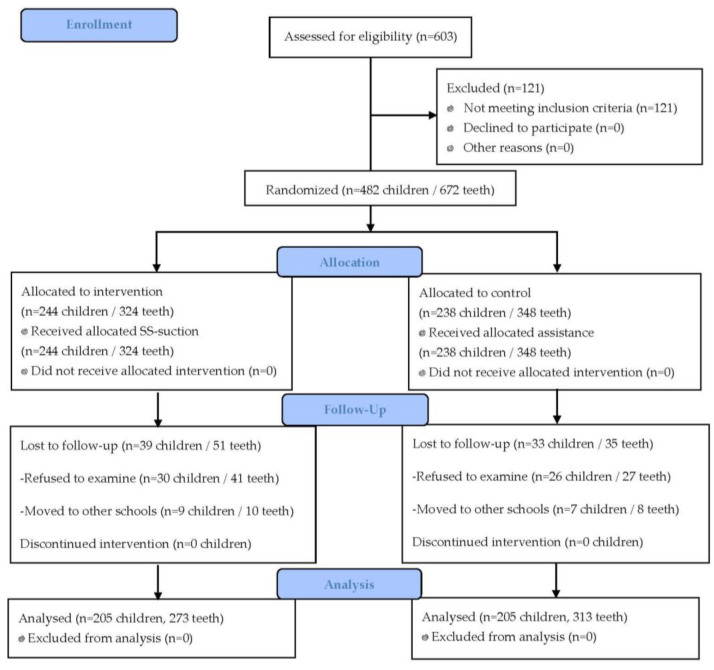
CONSORT Flow diagram.

**Table 1 ijerph-20-04035-t001:** Baseline characteristics of all cases (482 children and 672 teeth) by groups.

Characteristics	Moisture Control Type (%)		
	Control (DA) *	Intervention (SS) **	*p*-Value
** Child level (*n* = 482) **			
**Gender (*n* = 482)**			
Boy	120 (50.4)	117 (48.0)	0.588
Girl	118 (49.6)	127 (52.0)	
**Age (*n* = 482)**			
5–7 years	193 (81.1)	203 (83.2)	0.546
≥8 years	45 (18.9)	41 (16.8)	
**Tooth brushing (*n* = 461)**			
Regular (≥2 times per day)	86 (38.2)	111 (47.0)	0.056
Irregular (<2 times per day)	139 (61.8)	125 (53.0)	
**Fluoride toothpaste (*n* = 339)**			
Yes	161 (97.0)	169 (97.7)	0.689
No	5 (3.0)	4 (2.3)	
**Sugar snack or drink consumption (*n* = 456)**			
Rarely/seldom	42 (18.7)	38 (16.5)	0.596
Not everyday	117 (52.0)	131 (56.7)	
Everyday	66 (29.3)	62 (26.8)	
** Tooth level (*n* = 672 teeth) **			
**Child cooperation (*n* = 672)**			
Moderate	36 (10.3)	23 (7.1)	0.137
Good	312 (89.7)	301 (92.9)	
**Tooth position (*n* = 672)**			
Left	172 (49.4)	180 (55.6)	0.112
Right	176 (50.6)	144 (44.4)	

* Control = 238 children, 348 teeth; ** Intervention = 244 children, 324 teeth.

**Table 2 ijerph-20-04035-t002:** Baseline characteristics of follow-up cases (410 children 586 teeth) at 15–18 months by groups.

Characteristics	Moisture Control Type (%)		
	Control (DA) *	Intervention (SS) **	*p*-Value
** Child level (*n* = 410) **			
**Gender (*n* = 410)**			
Boy	107 (52.2)	98 (47.8)	0.374
Girl	98 (47.8)	107 (52.2)	
**Age (*n* = 410)**			
5–7 years	166 (81.0)	168 (82.0)	0.799
≥8 years	39 (19.0)	37 (18.0)	
**Tooth brushing (*n* = 396)**			
Regular (≥2 times per day)	75 (38.3)	92 (46.0)	0.119
Irregular (<2 times per day)	121 (61.7)	108 (54.0)	
**Fluoride toothpaste (*n* = 299)**			
Yes	142 (96.6)	149 (98.0)	0.444
No	5 (3.4)	3 (2.0)	
**Sugar snack or drink consumption (*n* = 391)**			
Rarely/seldom	38 (19.4)	34 (17.4)	0.686
Not everyday	100 (51.0)	108 (55.4)	
Everyday	58 (29.6)	53 (27.2)	
** Tooth level (*n* = 586 teeth) **			
**Child cooperation (*n* = 586)**			
Moderate	33 (10.5)	20 (7.3)	0.176
Good	280 (89.5)	253 (92.7)	
**Tooth position (*n* = 586)**			
Left	151 (48.8)	152 (55.5)	0.072
Right	162 (51.6)	121 (44.5)	
**Follow-up time (*n* = 586)**			<0.001**
15 months	133 (42.5)	182 (66.7)	
16 months	111 (35.5)	91 (33.3)	
17 and 18 months #	69 (22.0)	0 (0.0)	

* Control = 205 children, 313 teeth; ** Intervention = 205 children, 273 teeth; # FU at 18 months = 6 children, 9 teeth.

**Table 3 ijerph-20-04035-t003:** Satisfaction and safety toward SS-suction innovation of 15 dental nurses (244 children and 324 teeth).

Satisfaction	Teeth 36 (Left Side) (*n* = 180)				Teeth 46 (Right Side) (*n* = 144)			
	Median	Q1 *, Q3 *	Mean	S.D.	Median	Q1, Q3	Mean	S.D.
Tongue retraction	9	8, 10	8.94	1.15	9	8, 10	9.11	1.08
Buccal retraction	9	8, 10	9.04	1.09	9	8, 10	9.28	0.94
Insertion and removal	9	8, 10	8.63	1.37	9	8, 10	8.77	1.37
Attachment with the patient’s jaw	9	8, 10	8.67	1.20	9	8, 10	8.89	1.18
Obstructing working area	9	8, 10	8.96	1.04	9	8, 10	9.13	1.15
Suction saliva	9	8, 10	8.89	1.19	9	8, 10	8.98	1.25
Child cooperation	9	8, 10	8.77	1.44	9	8, 10	9.01	1.38
Overall	9	8, 10	8.91	1.02	9	8, 10	8.93	1.11
**Safety**	**Frequency (%) *n* = 180**				**Frequency (%) *n* = 144**			
Uncomfortable/Pain in mouth	30 (16.7)				26 (18.1)			
Uncomfortable/Pain at chin	12 (6.7)				12 (8.3)			
Ulcer in mouth **	2 (1.1)				2 (1.4)			
Ulcer at chin **	6 (3.3)				4 (2.8)			

* Q1 = 1st quartile and Q3 = 3rd quartile; ** Red area from pressure recovered after 5–10 min observation.

**Table 4 ijerph-20-04035-t004:** Oral hygiene and caries on sealed surfaces of follow-up teeth at 15–18 months by groups (410 children and 586 teeth).

Characteristics	Moisture Control Type (%)		
	Control (DA) *	Intervention (SS) **	*p*-Value
**Oral Hygiene (*n* = 583)** ^#^			
Good	125 (40.5)	115 (42.0)	0.710
Poor	184 (59.5)	159 (58.0)	
**Caries on sealed surface occlusal (*n* = 586)**			
Yes	86 (27.5)	73 (26.7)	0.853
No	227 (72.5)	200 (73.3)	
**Caries on sealed surface buccal (*n* = 586)**			
Yes	114 (36.4)	96 (35.2)	0.796
No	199 (63.6)	177 (64.8)	

* Control = 205 children, 313 teeth; ** Intervention = 205 children, 273 teeth; ^#^ three teeth were n/a due to large caries and loss of tooth surfaces.

**Table 5 ijerph-20-04035-t005:** Effect of SS-suction on occlusal caries controlling for confounding factors: result from multilevel logistic regression (FU cases).

Variables (Reference)	OR	95%CI	*p*-Value
** *Fixed effect* **			
Intercept	8.63	3.24, 22.97	<0.001 **
Tooth (Right vs. left)	0.80	0.55, 1.16	0.238
Group (SS-suction vs. control)	1.06	0.53, 2.13	0.861
Age (8 yrs+ vs. 5–7 years)	1.09	0.40, 3.00	0.862
Follow-up (T2 vs. T1)	0.40	0.15, 1.08	0.072
Follow-up (T3 vs. T1)	1.21	0.30, 4.91	0.795
**Random effect**: Standard deviation	Level 1 (Tooth) = 1 Level 2 (Children) = 2.471 Level 3 (Dental nurse) = 1.363		

Reference level for “caries on sealed surfaces” = Yes; T1= 15 months, T2 = 16 months, T3 = 17 and 18 months and * Significant level < 0.05 ****** Significant level < 0.001.

**Table 6 ijerph-20-04035-t006:** Effect of SS-suction on buccal caries controlling for confounding factors: result from multilevel logistic regression (FU cases).

Variables (Reference)	OR	95%CI	*p*-Value
** *Fixed effect* **			
Intercept	2.56	1.42, 4.61	0.002 *
Tooth (Right vs. left)	0.87	0.61, 1.23	0.419
Group (SS-suction vs. control)	1.07	0.63, 1.79	0.810
Age (8 yrs+ vs. 5-7 years)	1.47	0.73, 2.95	0.277
Follow-up (T2 vs. T1)	0.57	0.29, 1.11	0.102
Follow-up (T3 vs. T1)	0.90	0.35, 2.35	0.833
**Random effect**: Standard deviation	Level 1 (Tooth) = 1 Level 2 (Children) = 1.765 Level 3 (Dental nurse) = 0.577		

Reference level for “caries on sealed surfaces” = Yes; T1 (ref) = 15 months, T2 = 16 months, T3 = 17 and 18 months and * Significant level < 0.05 ** Significant level < 0.001.

## Data Availability

Petty patent number of 2-way SS-suction is 19211, the patent was registered under the authors’ names. The data that support the findings of this study are available from the corresponding author, on request.

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
