# Peer review of "A Cluster Randomized Controlled Trial of the Dental Sealants Quality in Rural Schoolchildren Using Innovative Suction without Dental Assistance"

_ijerph, 2023, doi:10.3390/ijerph20054035_

Round 1

Reviewer 1 Report

The authors have performed an amazing innovative concept.

Kudos to the authors.

Some comment for improvements of the paper

Please check the font size of the in text citation numbers they are larger than the brackets. Evaluate this through out the manuscript.

"Recently, the innovation (tongue and cheek retractor and saliva contamination (SS-suction))..." who have invented this? if other than the authors they should be acknowledged by citing their paper. Please do that.

In Figure 1 the authors should mark in the components, rather than putting this information in the figure legend. Please revise the figure.

In Figure 2 and Figure 3 the authors can mark the components that are seen in the clinical pictures.

The Figure 4 need to be according to the CONSORT 2010 flow diagram.

The petty patent should fall under the data availability or Acknowledgments.

Author Response

Dear the reviewer I,

         Thank you very much for your suggestion, we revised our manuscript as your suggestion.

         Best reagrds

         S Tianviwat

Reviewer 2 Report

Dear Authors,

Congratulations on the work you have done and presented in this manuscript. Unfortunately, for now, I cannot recommend publication in IJERPH, MDPI, as a high quality journal. 

First and foremost, there are multiple grammar and spelling errors, English needs to be revised. The methodology of your study is not conducted properly and the results are questionable. A quarter of the total number of the references are represented by your previous work. References section needs improvements. Please see the attachment.

Author Response

Dear Reviewer II,

         Thnak you very much for your suggestion, we revised the manuscript as your suggestions.

         Sincerely yours,

         S Tianviwat

Round 2

Reviewer 2 Report

Dear Authors,

I really appreciate the hard work you performed to improve the manuscript. Methodology is much more clearer now, images are improved a lot. I believe that your work has merits now for acceptance and it's ready for publication. I have  no further comments.

PS: check the style of last references added